# Sigsoftmax: Reanalysis of the Softmax Bottleneck

**Sekitoshi Kanai**
NTT Software Innovation Center, Keio Univ.
`kanai.sekitoshi@lab.ntt.co.jp`

**Yasuhiro Fujiwara**
NTT Software Innovation Center
`fujiwara.yasuhiro@lab.ntt.co.jp`

**Yuki Yamanaka**
NTT Secure Platform Laboratories
`yamanaka.yuki@lab.ntt.co.jp`

**Shuichi Adachi**
Keio Univ.
`adachi.shuichi@appi.keio.ac.jp`

## Abstract

Softmax is an output activation function for modeling categorical probability distributions in many applications of deep learning. However, a recent study revealed that softmax can be a bottleneck of representational capacity of neural networks in language modeling (the softmax bottleneck). In this paper, we propose an output activation function for breaking the softmax bottleneck without additional parameters. We re-analyze the softmax bottleneck from the perspective of the output set of log-softmax and identify the cause of the softmax bottleneck. On the basis of this analysis, we propose sigsoftmax, which is composed of a multiplication of an exponential function and sigmoid function. Sigsoftmax can break the softmax bottleneck. The experiments on language modeling demonstrate that sigsoftmax and mixture of sigsoftmax outperform softmax and mixture of softmax, respectively.

## 1  Introduction

Deep neural networks are used in many recent applications such as image recognition [17, 13], speech recognition [12], and natural language processing [24, 32, 7]. High representational capacity and generalization performance of deep neural networks are achieved by many layers, activation functions and regularization methods [26, 13, 31, 14, 10]. Although various model architectures are built in the above applications, softmax is commonly used as an output activation function for modeling categorical probability distributions [4, 10, 13, 24, 32, 7, 12]. For example, in language modeling, softmax is employed for representing the probability of the next word over the vocabulary in a sentence. When using softmax, we train the model by minimizing negative log-likelihood with a gradient-based optimization method. We can easily calculate the gradient of negative log-likelihood with softmax, and it is numerically stable [3, 4].

Even though softmax is widely used, few studies have attempted to improve its modeling performance [6, 8]. This is because deep neural networks with softmax are believed to have a universal approximation property. However, Yang et al. [34] recently revealed that softmax can be a bottleneck of representational capacity in language modeling. They showed that the representational capacity of the softmax-based model is restricted by the length of the hidden vector in the output layer. In language modeling, the length of the hidden vector is much smaller than the vocabulary size. As a result, the softmax-based model cannot completely learn the true probability distribution, and this is called the softmax bottleneck. For breaking the softmax bottleneck, Yang et al. [34] proposed mixture of softmax (MoS) that mixes the multiple softmax outputs. However, this analysis of softmax does not explicitly show why softmax can be a bottleneck. Furthermore, MoS is an additional layer or mixture model rather than an alternative activation function to softmax: MoS has learnable parameters and hyper-parameters.

In this paper, we propose a novel output activation function for breaking the softmax bottleneck without additional parameters. We re-analyze the softmax bottleneck from the point of view of the output set (range) of a function and show why softmax can be a bottleneck. This paper reveals that (i) the softmax bottleneck occurs because softmax uses only exponential functions for nonlinearity and (ii) the range of log-softmax is a subset of the vector space whose dimension depends on the dimension of the input space. As an alternative activation function to softmax, we explore the output functions composed of rectified linear unit (ReLU) and sigmoid functions. In addition, we propose *sigsoftmax*, which is composed of a multiplication of an exponential function and sigmoid function. Sigsoftmax has desirable properties for output activation functions, e.g., the calculation of its gradient is numerically stable. More importantly, sigsoftmax can break the softmax bottleneck, and the range of softmax can be a subset of that of sigsoftmax. Experiments on language modeling demonstrate that sigsoftmax can break the softmax bottleneck and outperform softmax. In addition, mixture of sigsoftmax outperforms MoS.

## 2 Preliminaries

### 2.1 Softmax

Deep neural networks use softmax in learning categorical distributions. For example, in the classification, a neural network uses softmax to learn the probability distribution over $M$ classes $\boldsymbol{y} \in \boldsymbol{R}^M$ conditioned on the input $\boldsymbol{x}$ as $P_{\boldsymbol{\theta}}(\boldsymbol{y}|\boldsymbol{x})$ where $\boldsymbol{\theta}$ is a parameter. Let $\boldsymbol{h}(\boldsymbol{x}) \in \boldsymbol{R}^d$ be a hidden vector and $\boldsymbol{W} \in \boldsymbol{R}^{M \times d}$ be a weight matrix in the output layer, the output of softmax $\boldsymbol{f}_s(\cdot)$ represents the conditional probability of the $i$-th class as follows:

$$P_{\boldsymbol{\theta}}(y_i|\boldsymbol{x}) = [\boldsymbol{f}_s(\boldsymbol{W}\boldsymbol{h}(\boldsymbol{x}))]_i = \frac{\exp([\boldsymbol{W}\boldsymbol{h}(\boldsymbol{x})]_i)}{\sum_{m=1}^{M} \exp([\boldsymbol{W}\boldsymbol{h}(\boldsymbol{x})]_m)}, \quad (1)$$

where $[\boldsymbol{f}_s]_i$ represents the $i$-th element of $\boldsymbol{f}_s$. We can see that each element of $\boldsymbol{f}_s$ is bounded from zero to one since the output of exponential functions is non-negative in eq. (1). The summation of all elements of $\boldsymbol{f}_s$ is obviously one. From these properties, we can regard output of the softmax trained by minimizing negative log-likelihood as a probability [4, 21]. If we only need the most likely label, we can find such a label by comparing elements of $\boldsymbol{W}\boldsymbol{h}(\boldsymbol{x})$ without the calculations of softmax $\boldsymbol{f}_s(\boldsymbol{W}\boldsymbol{h}(\boldsymbol{x}))$ once we have trained the softmax-based model. This is because exponential functions in softmax are monotonically increasing.

To train the softmax-based models, negative log-likelihood (cross entropy) is used as a loss function. Since the loss function is minimized by stochastic gradient descent (SGD), the properties of the gradients of functions are very important [26, 28, 9, 15]. One advantage of softmax is that the gradient of log-softmax is easily calculated as follows [3, 4, 1, 8]:

$$\frac{\partial [\log \boldsymbol{f}_s(\boldsymbol{z})]_i}{\partial z_j} = \begin{cases} 1 - [\boldsymbol{f}_s(\boldsymbol{z})]_j & \text{if } j = i, \\ -[\boldsymbol{f}_s(\boldsymbol{z})]_j & \text{if } j \neq i, \end{cases} \quad (2)$$

where $\boldsymbol{z} = \boldsymbol{W}\boldsymbol{h}(\boldsymbol{x})$. Whereas the derivative of the logarithm can cause a division by zero since $\frac{\mathrm{d}\log(z)}{\mathrm{d}z} = \frac{1}{z}$, the derivative of log-softmax cannot. As a result, softmax is numerically stable.

### 2.2 Softmax bottleneck

In recurrent neural network (RNN) language modeling, given a corpus of tokens $\boldsymbol{Y} = (Y_1, \ldots, Y_T)$, the joint probability $P(\boldsymbol{Y})$ is factorized as $P(\boldsymbol{Y}) = \prod_t P(Y_t|Y_{<t}) = \prod_t P(Y_t|X_t)$, where $X_t = Y_{<t}$ is referred to as the context of the conditional probability. Output of softmax $\boldsymbol{f}_s(\boldsymbol{W}\boldsymbol{h}(X_t))$ learns $P(Y_t|X_t)$ where (a) $\boldsymbol{h}(X_t) \in \boldsymbol{R}^d$ is the hidden vector corresponding to the context $X_t$ and (b) $\boldsymbol{W}$ is a weight matrix in the output layer (embedding layer). A natural language is assumed as a finite set of pairs of $x_t$ and $P^*(Y|x_t)$ as $\mathcal{L} = \{(x_1, P^*(Y|x_1)), \ldots, (x_N, P^*(Y|x_N))\}$, where $N$ is the number of possible contexts. The objective of language modeling is to learn a model distribution $P_{\boldsymbol{\theta}}(Y|X)$ parameterized by $\boldsymbol{\theta}$ to match the true data distribution $P^*(Y|X)$. Note that upper- and lower-case letters are used for variables and constants, respectively, in this section. Under the above assumptions, let $y_1, \ldots, y_M$ be $M$ possible tokens in the language $\mathcal{L}$, the previous study of Yang

et al. [34] considers the following three matrices:

$$
\boldsymbol{H_\theta} = \begin{bmatrix} \boldsymbol{h}(x_1)^T \\ \boldsymbol{h}(x_2)^T \\ \vdots \\ \boldsymbol{h}(x_N)^T \end{bmatrix}, \boldsymbol{W}, \boldsymbol{A} = \begin{bmatrix} \log P^*(y_1|x_1), & \log P^*(y_2|x_1), & \dots & \log P^*(y_M|x_1) \\ \log P^*(y_1|x_2), & \log P^*(y_2|x_2), & \dots & \log P^*(y_M|x_2) \\ \vdots & \vdots & \ddots & \vdots \\ \log P^*(y_1|x_N), & \log P^*(y_2|x_N), & \dots & \log P^*(y_M|x_N) \end{bmatrix}. \quad (3)
$$

$\boldsymbol{H_\theta} \in \boldsymbol{R}^{N \times d}$ is a matrix composed of the hidden vectors, $\boldsymbol{W} \in \boldsymbol{R}^{M \times d}$ is a weight matrix, and $\boldsymbol{A} \in \boldsymbol{R}^{M \times N}$ is a matrix composed of the log probabilities of the true distribution. By using these matrices, the rank of $\boldsymbol{H_\theta} \boldsymbol{W}^T$ should be greater than or equal to $\operatorname{rank}(\boldsymbol{A}) - 1$ so that the softmax-based model completely learns $\mathcal{L}$ [34]. However, the rank of $\boldsymbol{H_\theta} \boldsymbol{W}^T$ is at most $d$ if any functions $\mathcal{U}$ are used for $\boldsymbol{H_\theta}$ and $\boldsymbol{W}$. Therefore, if we have $d < \operatorname{rank}(\boldsymbol{A}) - 1$, softmax can be the bottleneck of representational capacity as shown in the following theorem:

**Theorem 1** (Softmax Bottleneck [34])**.** *If $d < \operatorname{rank}(\boldsymbol{A}) - 1$, for any function family $\mathcal{U}$ and any model parameter $\boldsymbol{\theta}$, there exists a context $x$ in $\mathcal{L}$ such that $P_{\boldsymbol{\theta}}(Y|x) \neq P^*(Y|x)$.*

This theorem shows that the length of the hidden vector in the output layer determines the representational power of RNN with softmax. In language modeling, the rank of $\boldsymbol{A}$ can be extremely high since contexts can vary and vocabulary size $M$ is much larger than $d$. Therefore, the softmax can be the bottleneck of the representational power.

## 2.3 Mixture of softmax

A simple approach to improving the representational capacity is to use a weighted sum of the several models. In fact, Yang et al. [34] use this approach for breaking the softmax bottleneck. As the alternative to softmax, they propose the mixture of softmax (MoS), which is the weighted sum of $K$ softmax functions:

$$
P_{\boldsymbol{\theta}}(y_i|x) = \sum_{k=1}^{K} \pi(x,k) \frac{\exp([\boldsymbol{W}\boldsymbol{h}(x,k)]_i)}{\sum_{m=1}^{M} \exp([\boldsymbol{W}\boldsymbol{h}(x,k)]_m)}, \quad (4)
$$

where $\pi(x,k)$ is the prior or mixture weight of the $k$-th component, and $\boldsymbol{h}(x,k)$ is the $k$-th context vector associated with the context $x$. Let $\boldsymbol{h}'(x)$ be input of MoS for the context $x$. The priors and context vectors are parameterized as $\pi(x,k) = \frac{\exp(\boldsymbol{w}_{\pi,k}^T \boldsymbol{h}'(x))}{\sum_{k'=1}^{K} \exp(\boldsymbol{w}_{\pi,k'}^T \boldsymbol{h}'(x))}$ and $\boldsymbol{h}(x,k) = \tanh(\boldsymbol{W}_{h,k}\boldsymbol{h}'(x))$, respectively. MoS can break the softmax bottleneck since the rank of the approximate $\boldsymbol{A}$ can be arbitrarily large [34]. Therefore, language modeling with MoS performs better than that with softmax. However, in this method, the number of mixtures $K$ is the hyper-parameter which needs to be tuned. In addition, weights $\boldsymbol{W}_{h,k}$ and $\boldsymbol{w}_{\pi,k}$ are additional parameters. Thus, MoS can be regarded as an additional layer or mixing technique rather than the improvement of the activation function.

## 2.4 Related work

Previous studies proposed alternative functions to softmax [8, 25, 27]. The study of de Brébisson and Vincent [8] explored spherical family functions: the spherical softmax and Taylor softmax. They showed that these functions do not outperform softmax when the length of an output vector is large. In addition, the spherical softmax has a hyper-parameter that should be carefully tuned for numerical stability reasons [8]. On the other hand, the Taylor softmax might suffer from the softmax bottleneck since it approximates softmax. Mohassel and Zhang [25] proposed a ReLU-based alternative function to softmax for privacy-preserving machine learning since softmax is expensive to compute inside a secure computation. However, it leads to a division by zero since all outputs of ReLUs frequently become zeros and the denominator for normalization becomes zero. Several studies improved the efficiency of softmax [11, 30, 33, 20]. However, they did not improve the representational capacity.

# 3 Proposed method

## 3.1 Reanalysis of the softmax bottleneck

The analysis of the softmax bottleneck [34] is based on matrix factorization and reveals that the rank of $\boldsymbol{H_\theta} \boldsymbol{W}_\theta^T$ needs to be greater than or equal to $\operatorname{rank}(A) - 1$. Since the rank of $\boldsymbol{H_\theta} \boldsymbol{W}_\theta^T$ becomes

the length of the hidden vector in the output layer, the length of the hidden vector determines the representational power as described in Sec. 2.2. However, this analysis does not explicitly reveal the cause of the softmax bottleneck. To identify the cause of the softmax bottleneck, we re-analyze the softmax bottleneck from the perspective of the range of log-softmax because it should be large enough to approximate the true log probabilities.

Log-softmax is a logarithm of softmax and is used in training of deep learning as mentioned in Sec. 2.1. By using the notation in Sec. 2.1, log-softmax $\log(\boldsymbol{f}_s(\boldsymbol{z}))$ can be represented as $[\log(\boldsymbol{f}_s(\boldsymbol{z}))]_i = \log\left(\frac{\exp(z_i)}{\sum_{m=1}^{M}\exp(z_m)}\right) = z_i - \log(\sum_{m=1}^{M}\exp(z_m))$. This function can be expressed as

$$\log(\boldsymbol{f}_s(\boldsymbol{z})) = \boldsymbol{z} - \log(\textstyle\sum_{m=1}^{M}\exp(z_m))\mathbf{1}, \tag{5}$$

where $\mathbf{1}$ is the vector of all ones. To represent various log probability distributions $\log(P^*(\boldsymbol{y}|\boldsymbol{x}))$, the range of $\log(\boldsymbol{f}_s(\boldsymbol{z})) \in \boldsymbol{R}^M$ should be sufficiently large. Therefore, we investigate the range of $\log(\boldsymbol{f}_s(\boldsymbol{z}))$. We assume that the hidden vector $\boldsymbol{h}$ in the output layer can be an arbitrary vector in $\boldsymbol{R}^d$ where $d \leq M$, and the weight matrix $\boldsymbol{W} \in \boldsymbol{R}^{M \times d}$ is the full rank matrix; the rank of $\boldsymbol{W}$ is $d$.[1] Under these assumptions, the input vector space of softmax $S$ ($\boldsymbol{z} \in S$) is a $d$ dimensional vector space, and we have the following theorem:

**Theorem 2.** *Let $S \subseteq \boldsymbol{R}^M$ be the $d$ dimensional vector space and $\boldsymbol{z} \in S$ be input of log-softmax, every range of the log-softmax $\{\log(\boldsymbol{f}_s(\boldsymbol{z}))|\boldsymbol{z} \in S\}$ is a subset of the $d+1$ dimensional vector space.*

*Proof.* The input of log-softmax $\boldsymbol{z} = \boldsymbol{W}\boldsymbol{h}$ can be represented by $d$ singular vectors of $\boldsymbol{W}$ since the rank of $\boldsymbol{W}$ is $d$. In other words, the space of input vectors $\boldsymbol{z}$ is spanned by $d$ basis vectors. Thus, the input vector space $\{\boldsymbol{z}|\boldsymbol{z} \in S\}$ is represented as $\{\sum_{l=1}^{d}k^{(l)}\boldsymbol{u}^{(l)}|k^{(l)} \in \boldsymbol{R}\}$ where $\boldsymbol{u}^{(l)} \in \boldsymbol{R}^M$ for $l = 1, \ldots, d$ are linearly independent vectors and $k^{(l)}$ are their coefficients. From eq. (5), by using $\boldsymbol{u}^{(l)}$ and $k^{(l)}$, the range of log-softmax $\{\log(\boldsymbol{f}_s(\boldsymbol{z}))|\boldsymbol{z} \in S\}$ becomes

$$\{\log(\boldsymbol{f}_s(\boldsymbol{z}))|\ \boldsymbol{z} \in S\} = \{\textstyle\sum_{l=1}^{d}k^{(l)}\boldsymbol{u}^{(l)} - c(\sum_{l=1}^{d}k^{(l)}\boldsymbol{u}^{(l)})\mathbf{1}|k^{(l)} \in \boldsymbol{R}\}, \tag{6}$$

where $c(\sum_{l=1}^{d}k^{(l)}\boldsymbol{u}^{(l)}) = \log(\sum_{m=1}^{M}\exp(\left[\sum_{l=1}^{d}k^{(l)}\boldsymbol{u}^{(l)}\right]_m))$. This is the linear combination of $d$ linearly independent vectors $\boldsymbol{u}^{(l)}$ and $\mathbf{1}$. Therefore, we have the following relation:

$$\{\textstyle\sum_{l=1}^{d}k^{(l)}\boldsymbol{u}^{(l)} - c(\sum_{l=1}^{d}k^{(l)}\boldsymbol{u}^{(l)})\mathbf{1}|k^{(l)} \in \boldsymbol{R}\} \subseteq \{\sum_{l=1}^{d}k^{(l)}\boldsymbol{u}^{(l)} + k^{(d+1)}\mathbf{1}|k^{(l)} \in \boldsymbol{R}\}, \tag{7}$$

where $\{\sum_{l=1}^{d}k^{(l)}\boldsymbol{u}^{(l)} + k^{(d+1)}\mathbf{1}|k^{(l)} \in \boldsymbol{R}\}$ is the vector space spanned by $\boldsymbol{u}^{(l)}$ and $\mathbf{1}$. Let $Y$ be the vector space $\{\sum_{l=1}^{d}k^{(l)}\boldsymbol{u}^{(l)} + k^{(d+1)}\mathbf{1}|k^{(l)} \in \boldsymbol{R}\}$, the dimension of $Y$ becomes

$$\dim(Y) = \begin{cases} d+1 & \text{if } \mathbf{1} \notin \{\sum_{l=1}^{d}k^{(l)}\boldsymbol{u}^{(l)}|k^{(l)} \in \boldsymbol{R}\}, \\ d & \text{if } \mathbf{1} \in \{\sum_{l=1}^{d}k^{(l)}\boldsymbol{u}^{(l)}|k^{(l)} \in \boldsymbol{R}\}. \end{cases} \tag{8}$$

We can see that $Y$ is the $d$ or $d+1$ dimensional linear subspace of $\boldsymbol{R}^M$. From eqs. (7) and (8), output vectors of log-softmax exist in the $d+1$ dimensional vector space, which completes the proof. □

Theorem 2 shows that the log-softmax has at most $d+1$ linearly independent output vectors, even if the various inputs are applied to the model. Therefore, if the vectors of true log probabilities $\log P^*(\boldsymbol{y}|\boldsymbol{x})$ have more than $d+1$ linearly independent vectors, the softmax-based model cannot completely represent the true probabilities. Figure 1 illustrates theorems 1 and 2 when $M = 3$ and $d = 1$. We can prove Theorem 1 by using Theorem 2 as follows:

*Proof.* If we have $d < \text{rank}(\boldsymbol{A}) - 1$, i.e., $\text{rank}(\boldsymbol{A}) > d+1$, the number of linearly independent vectors of $\log P^*(\boldsymbol{y}|\boldsymbol{x})$ is larger than $d+1$. On the other hand, the output vectors $\log P_{\boldsymbol{\theta}}(\boldsymbol{y}|\boldsymbol{x})$ of the model cannot be larger than $d+1$ linearly independent vectors from Theorem 2. Therefore, the softmax-based model cannot completely learn $P^*(\boldsymbol{y}|\boldsymbol{x})$, i.e., there exists a context $x$ in $\mathcal{L}$ such that $P_{\boldsymbol{\theta}}(Y|x) \neq P^*(Y|x)$. □

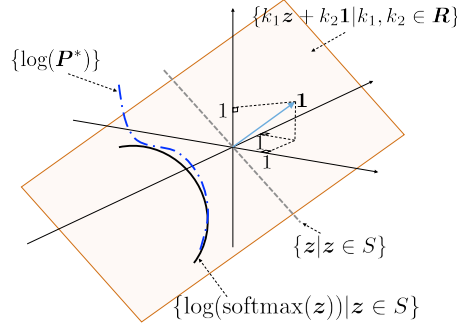

Figure 1: Softmax bottleneck ($M = 3$, $d = 1$). The input space $S$ is a gray dashed straight line, and the range of log-softmax is the black curve on the orange plane spanned by $\boldsymbol{z}$ and $\mathbf{1}$. On the other hand, $\{\log(\boldsymbol{P}^*)\}$ is the blue dash-dotted curve over the 3 dimensional (3-D) space since $M=3$. We can see that the range of log-softmax cannot match the $\{\log(\boldsymbol{P}^*)\}$ over the 3-D space.

The above analysis shows that the softmax bottleneck occurs because the output of log-softmax is the linear combination of the input $\boldsymbol{z}$ and vector $\mathbf{1}$ as eq. (5). Linear combination of the input and vector $\mathbf{1}$ increases the number of linearly independent vectors by at most one, and as a result, the output vectors become at most $d + 1$ linearly independent vectors. The reason log-softmax becomes the linear combination is that the logarithm of the exponential function $\log(\exp(z))$ is $z$.

By contrast, the number of linearly independent output vectors of a nonlinear function can be much greater than the number of linearly independent input vectors. Therefore, if the other nonlinear functions are replaced with exponential functions, the logarithm of such functions can be nonlinear and the softmax bottleneck can be broken without additional parameters.

Our analysis provides new insights that the range of log-softmax is a subset of the less dimensional vector space although the dimension of a vector space is strongly related to the rank of a matrix. Furthermore, our analysis explicitly shows the cause of the softmax bottleneck.

### 3.2 Alternative functions to softmax and desirable properties

In the previous section, we explained that the softmax bottleneck can be broken by replacing nonlinear functions with exponential functions. In this section, we explain the desirable properties of an alternative function to softmax. We formulate a new output function $\boldsymbol{f}(\cdot)$ as follows:

$$[\boldsymbol{f}(\boldsymbol{z})]_i = \frac{[\boldsymbol{g}(\boldsymbol{z})]_i}{\sum_{m=1}^{M}[\boldsymbol{g}(\boldsymbol{z})]_m}. \tag{9}$$

The new function is composed of the nonlinear function $\boldsymbol{g}(\boldsymbol{z})$ and the division for the normalization so that the summation of the elements is one. As the alternative function to softmax, a new output function $\boldsymbol{f}(\boldsymbol{z})$ and its $\boldsymbol{g}(\boldsymbol{z})$ should have all of the following properties:

**Nonlinearity of** $\log(\boldsymbol{g}(\boldsymbol{z}))$ As mentioned in Secs. 2.2 and 3.1, softmax can be the bottleneck of the representational power because $\log(\exp(\boldsymbol{z}))$ is $\boldsymbol{z}$. Provided that $\log(\boldsymbol{g}(\boldsymbol{z}))$ is a linear function, $\{\log(\boldsymbol{f}(\boldsymbol{z}))|\boldsymbol{z} \in S\}$ is a subset of the $d + 1$ dimensional vector space. In order to break the softmax bottleneck, $\log(\boldsymbol{g}(\boldsymbol{z}))$ should be nonlinear.

**Numerically stable** In training of deep learning, we need to calculate the gradient for optimization. The derivative of logarithm of $[\boldsymbol{f}(\boldsymbol{z})]_i$ with respect to $z_j$ is

$$\frac{\partial \log([\boldsymbol{f}(\boldsymbol{z})]_i)}{\partial z_j} = \frac{1}{[\boldsymbol{f}(\boldsymbol{z})]_i}\frac{\partial[\boldsymbol{f}(\boldsymbol{z})]_i}{\partial z_j}. \tag{10}$$

We can see that this function has a division by $[\boldsymbol{f}(\boldsymbol{z})]_i$. It can cause a division by zero since $[\boldsymbol{f}(\boldsymbol{z})]_i$ can be close to zero if networks completely go wrong in training. The alternative functions should avoid a division by zero similar to softmax as shown in eq. (2).

**Non-negative** In eq. (9), all elements of $\boldsymbol{g}(\boldsymbol{z})$ should be non-negative to limit output in $[0, 1]$. Therefore, $\boldsymbol{g}(\boldsymbol{z})$ should be non-negative: $[\boldsymbol{g}(\boldsymbol{z})]_i \geq 0$. Note that if $\boldsymbol{g}(\boldsymbol{z})$ is non-positive, $\boldsymbol{f}(\boldsymbol{z})$ are also limited to $[0, 1]$. We only mention non-negative since non-positive functions $\boldsymbol{g}(\boldsymbol{z})$ can easily be non-negative as $-\boldsymbol{g}(\boldsymbol{z})$.

**Monotonically increasing** $g(z)$ should be monotonically increasing so that $f(z)$ becomes a smoothed version of the argmax function [4, 2]. If $g(z)$ is monotonically increasing, we can obtain the label that has the maximum value of $f(z)$ by comparing elements of $z$.

Note that, if we use ReLU as $g(z)$, the ReLU-based function $f(z)$ does not have all the above properties since the gradient of its logarithm is not numerically stable. If we use sigmoid as $g(z)$, the new sigmoid-based function satisfies the above properties. However, the output of sigmoid is bounded above as $[g(z)]_i \leq 1$, and this restriction might limit the representational power. In fact, the sigmoid-based function does not outperform softmax on the large dataset in Sec. 4. We discuss these functions in detail in the supplementary material. In the next section, we propose a new output activation function that can break the softmax bottleneck, and satisfies all the above properties.

### 3.3 Sigsoftmax

For breaking the softmax bottleneck, we propose sigsoftmax given as follows:

**Definition 1.** *Sigsoftmax is defined as*

$$[f(z)]_i = \frac{\exp(z_i)\sigma(z_i)}{\sum_{m=1}^{M} \exp(z_m)\sigma(z_m)}, \tag{11}$$

*where $\sigma(\cdot)$ represents a sigmoid function.*

We theoretically show that sigsoftmax can break the softmax bottleneck and has the desired properties. In the same way as in the analysis of softmax in Sec. 3.1, we examine the range of log-sigsoftmax. Since we have $\log(\sigma(z)) = \log(\frac{1}{1+\exp(-z)}) = z - \log(1+\exp(z))$, log-sigsoftmax becomes

$$\log(f(z)) = 2z - \log(1 + \exp(z)) + c'(z)\mathbf{1}, \tag{12}$$

where $c'(z) = \log(\sum_{m=1}^{M} \exp(z_m)\sigma(z_m))$, and $\log(1 + \exp(z))$ is the nonlinear function called softplus [10]. Since log-sigsoftmax is composed of a nonlinear function, its output vectors can be greater than $d + 1$ linearly independent vectors. Therefore, we have the following theorem:

**Theorem 3.** *Let $S \subseteq \mathbf{R}^M$ be the d dimensional vector space and $z \in S$ be input of log-sigsoftmax, some range of log-sigsoftmax $\{\log(f(z)) | z \in S\}$ is not a subset of a $d + 1$ dimensional vector space.*

The detailed proof of this theorem is given in the supplementary material. Theorem 3 shows that sigsoftmax can break the softmax bottleneck; even if the vectors of the true log probabilities are more than $d + 1$ linearly independent vectors, the sigsoftmax-based model can learn the true probabilities.

However, the representational powers of sigsoftmax and softmax are difficult to compare only by using the theorem based on the vector space. This is because both functions are nonlinear and their ranges are not necessarily vector spaces, even though they are subsets of vector spaces. Therefore, we directly compare the ranges of sigsoftmax and softmax as the following theorem:

**Theorem 4.** *Let $z \in S$ be the input of sigsoftmax $f(\cdot)$ and softmax $f_s(\cdot)$. If the $S$ is a d dimensional vector space and $\mathbf{1} \in S$, the range of softmax is a subset of the range of sigsoftmax*

$$\{f_s(z) | z \in S\} \subseteq \{f(z) | z \in S\}. \tag{13}$$

*Proof.* If we have $\mathbf{1} \in S$, $S$ can be written as $S = \{\sum_{l=1}^{d-1} k'^{(l)}\mathbf{u}'^{(l)} + k'^{(d)}\mathbf{1} | k'^{(l)} \in R\}$ where $\mathbf{u}'^{(l)}$ $(l = 1, \ldots, d-1)$ and $\mathbf{1}$ are linearly independent vectors. In addition, the arbitrary elements of $S$ can be written as $\sum_{l=1}^{d-1} k'^{(l)}\mathbf{u}'^{(l)} + k'^{(d)}\mathbf{1}$, and thus, $z = \sum_{l=1}^{d-1} k'^{(l)}\mathbf{u}'^{(l)} + k'^{(d)}\mathbf{1}$. For the output of softmax, by substituting $z = \sum_{l=1}^{d-1} k'^{(l)}\mathbf{u}'^{(l)} + k'^{(d)}\mathbf{1}$ for eq. (1), we have

$$[f_s(z)]_i = \frac{\exp([\sum_{l=1}^{d-1} k'^{(l)}\mathbf{u}'^{(l)}]_i + k'^{(d)})}{\sum_{m=1}^{M} \exp([\sum_{l=1}^{d-1} k'^{(l)}\mathbf{u}'^{(l)}]_m + k'^{(d)})} = \frac{\exp([\sum_{l=1}^{d-1} k'^{(l)}\mathbf{u}'^{(l)}]_i)}{\sum_{m=1}^{M} \exp([\sum_{l=1}^{d-1} k'^{(l)}\mathbf{u}'^{(l)}]_m)}. \tag{14}$$

As a result, the range of softmax becomes as follows:

$$\left\{ f_s(\textstyle\sum_{l=1}^{d-1} k'^{(l)}\mathbf{u}'^{(l)} + k'^{(d)}\mathbf{1}) | k'^{(l)} \in \mathbf{R} \right\} = \left\{ \frac{\exp([\sum_{l=1}^{d-1} k'^{(l)}\mathbf{u}'^{(l)}]_i)}{\sum_{m=1}^{M} \exp([\sum_{l=1}^{d-1} k'^{(l)}\mathbf{u}'^{(l)}]_m)} | k'^{(l)} \in \mathbf{R} \right\}. \tag{15}$$

On the other hand, by substituting $\boldsymbol{z} = \sum_{l=1}^{d-1} k'^{(l)} \boldsymbol{u}'^{(l)} + k'^{(d)} \boldsymbol{1}$ for eq. (11), the output of sigsoftmax becomes as follows:

$$[\boldsymbol{f}(\boldsymbol{z})]_i = \frac{\exp\left(\left[\sum_{l=1}^{d-1} k'^{(l)} \boldsymbol{u}'^{(l)}\right]_i\right)\sigma\left(\left[\sum_{l=1}^{d-1} k'^{(l)} \boldsymbol{u}'^{(l)}\right]_i + k'^{(d)}\right)}{\sum_{m=1}^{M} \exp\left(\left[\sum_{l=1}^{d-1} k'^{(l)} \boldsymbol{u}'^{(l)}\right]_m\right)\sigma\left(\left[\sum_{l=1}^{d-1} k'^{(l)} \boldsymbol{u}'^{(l)}\right]_m + k'^{(d)}\right)}. \tag{16}$$

When $k'^{(l)}$ are fixed for $l = 1, \ldots, d-1$ and $k'^{(d)} \to +\infty$,[2] we have the following equality:

$$\lim_{k'^{(d)} \to +\infty} \frac{\exp\left(\left[\sum_{l=1}^{d-1} k'^{(l)} \boldsymbol{u}'^{(l)}\right]_i\right)\sigma\left(\left[\sum_{l=1}^{d-1} k'^{(l)} \boldsymbol{u}'^{(l)}\right]_i + k'^{(d)}\right)}{\sum_{m=1}^{M} \exp\left(\left[\sum_{l=1}^{d-1} k'^{(l)} \boldsymbol{u}'^{(l)}\right]_m\right)\sigma\left(\left[\sum_{l=1}^{d-1} k'^{(l)} \boldsymbol{u}'^{(l)}\right]_m + k'^{(d)}\right)} = \frac{\exp\left(\left[\sum_{l=1}^{d-1} k'^{(l)} \boldsymbol{u}'^{(l)}\right]_i\right)}{\sum_{m=1}^{M} \exp\left(\left[\sum_{l=1}^{d-1} k'^{(l)} \boldsymbol{u}'^{(l)}\right]_m\right)}, \tag{17}$$

since $\lim_{k \to +\infty} \sigma(v+k) = 1$ when $v$ is fixed. From eq. (17), sigsoftmax has the following relation:

$$\left\{ \frac{\exp\left(\left[\sum_{l=1}^{d-1} k'^{(l)} \boldsymbol{u}'^{(l)}\right]_i\right)}{\sum_{m=1}^{M} \exp\left(\left[\sum_{l=1}^{d-1} k'^{(l)} \boldsymbol{u}'^{(l)}\right]_m\right)} | k'^{(l)} \in \boldsymbol{R} \right\} = \{\boldsymbol{f}(\boldsymbol{z})|\boldsymbol{z} \in S'\} \subseteq \{\boldsymbol{f}(\boldsymbol{z})|\boldsymbol{z} \in S\}, \tag{18}$$

where $S'$ is a hyperplane of $S$ with $k'^{(d)} = +\infty$, $S' = \{\sum_{l=1}^{d-1} k'^{(l)} \boldsymbol{u}'^{(l)} + k'^{(d)} \boldsymbol{1} | k'^{(l)} \in \boldsymbol{R}$ for $l = 1, \ldots, d-1, k'^{(d)} = +\infty\} \subset S$. From eqs. (15) and (18), we can see that the range of sigsoftmax includes the range of softmax. Therefore, we have $\{\boldsymbol{f}_s(\boldsymbol{z})|\boldsymbol{z} \in S\} \subseteq \{\boldsymbol{f}(\boldsymbol{z})|\boldsymbol{z} \in S\}$. $\qquad\square$

Theorem 4 shows that the range of sigsoftmax can be larger than that of softmax if $\boldsymbol{1} \in S$. The assumption $\boldsymbol{1} \in S$ means that there exist inputs of which outputs are the equal probabilities for all labels as $p_{\boldsymbol{\theta}}(y_i|\boldsymbol{x}) = \frac{1}{M}$ for all $i$. This assumption is not very strong in practice. If $\boldsymbol{1} \notin S$, the range of sigsoftmax can include the range of softmax by introducing one learnable scalar parameter $b$ into sigsoftmax as $[\boldsymbol{f}(\boldsymbol{z} + b\boldsymbol{1})]_i = \frac{\exp(z_i)\sigma(z_i+b)}{\sum_{m=1}^{M} \exp(z_m)\sigma(z_m+b)}$. In this case, if softmax can fit the true probability, $b$ can become large enough for sigsoftmax to approximately equal softmax. In the experiments, we did not use $b$ in order to confirm that sigsoftmax can outperform softmax without additional parameters. From Theorems 3 and 4, sigsoftmax can break the softmax bottleneck, and furthermore, the representational power of sigsoftmax can be higher than that of softmax.

Then, we show that sigsoftmax has the desirable properties introduced in Sec. 3.2 as shown in the following theorem from Definition 1 although we show its proof in the supplementary material:

**Theorem 5.** *Sigsoftmax has the following properties:*

1. *Nonlinearity of* $\log(\boldsymbol{g}(\boldsymbol{z}))$: $\log(\boldsymbol{g}(\boldsymbol{z})) = 2\boldsymbol{z} - \log(\boldsymbol{1} + \exp(\boldsymbol{z}))$.

2. *Numerically stable:* $\frac{\partial \log[\boldsymbol{f}(\boldsymbol{z})]_i}{\partial z_j} = \begin{cases} (1 - [\boldsymbol{f}(\boldsymbol{z})]_j)(2 - \sigma(z_j)) & i = j, \\ -[\boldsymbol{f}(\boldsymbol{z})]_j (2 - \sigma(z_j)) & i \neq j. \end{cases}$

3. *Non-negative:* $[\boldsymbol{g}(\boldsymbol{z})]_i = \exp(z_i)\sigma(z_i) \geq 0$.

4. *Monotonically increasing:* $z_1 \leq z_2 \Rightarrow \exp(z_1)\sigma(z_1) \leq \exp(z_2)\sigma(z_2)$.

Since sigsoftmax is an alternative function to softmax, we can use the weighted sum of sigsoftmax functions in the same way as MoS. Mixture of sigsoftmax (MoSS) is the following function:

$$P_{\boldsymbol{\theta}}(y_i|x) = \sum_{k=1}^{K} \pi(x,k) \frac{\exp([\boldsymbol{W}\boldsymbol{h}(x,k)]_i)\sigma([\boldsymbol{W}\boldsymbol{h}(x,k)]_i)}{\sum_{m=1}^{M} \exp([\boldsymbol{W}\boldsymbol{h}(x,k)]_m)\sigma([\boldsymbol{W}\boldsymbol{h}(x,k)]_m)}. \tag{19}$$

$\pi(x,k)$ is also composed of sigsoftmax as $\pi(x,k) = \frac{\exp(\boldsymbol{w}_{\pi,k}^T \boldsymbol{h}'(x))\sigma(\boldsymbol{w}_{\pi,k}^T \boldsymbol{h}'(x))}{\sum_{k'=1}^{K} \exp(\boldsymbol{w}_{\pi,k'}^T \boldsymbol{h}'(x))\sigma(\boldsymbol{w}_{\pi,k'}^T \boldsymbol{h}'(x))}$.

## 4 Experiments

To evaluate the effectiveness of sigsoftmax, we conducted experiments on word-level language modeling. We compared sigsoftmax with softmax, the ReLU-based function and the sigmoid-based function. We also compared the mixture of sigsoftmax with that of softmax; MoSS with MoS.

Note that we provide the character-level language modeling experiments on text8 [18] and word-level language modeling experiments on One Billion Word dataset [5] in the supplementary material. Since the softmax bottleneck does not occur on character-level language modeling, we confirmed the performance of sigsoftmax is similar to that of softmax in these experiments. On One Billion Word dataset, we used efficient method [11] since One Billion Word is the massive dataset. We confirmed that sigsoftmax can outperform softmax on the massive dataset.

Table 1: Results of the language modeling experiment on PTB.

|  | Softmax | $g$:ReLU | $g$: Sigmoid | Sigsoftmax | MoS | MoSS |
|---|---|---|---|---|---|---|
| Validation | 51.2±0.5 | $(4.91\pm5)\times10^3$ | **49.2±0.4** | 49.7±0.5 | 48.6±0.2 | **48.3±0.1** |
| Test | 50.5±0.5 | $(2.78\pm8)\times10^5$ | **48.9±0.3** | 49.2±0.4 | 48.0±0.1 | **47.7±0.07** |

Table 2: Results of the language modeling experiment on WT2.

|  | Softmax | $g$:ReLU | $g$:Sigmoid | Sigsoftmax | MoS | MoSS |
|---|---|---|---|---|---|---|
| Validation | 45.3±0.2 | $(1.79\pm0.8)\times10^3$ | 45.7±0.1 | **44.9±0.1** | 42.5±0.1 | **42.1±0.2** |
| Test | 43.3±0.1 | $(2.30\pm2)\times10^4$ | 43.5±0.1 | **42.9±0.1** | 40.8±0.03 | **40.3±0.2** |

## 4.1 Experimental conditions

We used Penn Treebank dataset (PTB) [19, 24] and WikiText-2 dataset (WT2) [22] by following the previous studies [23, 16, 34]. PTB is commonly used to evaluate the performance of RNN-based language modeling [24, 35, 23, 34]. PTB is split into a training set (about 930 k tokens), validation set (about 74 k tokens), and test set (about 82 k tokens). The vocabulary size $M$ was set to 10 k, and all words outside the vocabulary were replaced with a special token. WT2 is a collection of tokens from the set of articles on Wikipedia. WT2 is also split into a training set (about 2100 k), validation set (about 220 k), and test set (about 250 k). The vocabulary size $M$ was 33,278. Since WT2 is larger than PTB, language modeling of WT2 may require more representational power than that of PTB.

We trained a three-layer long short-term memory (LSTM) model with each output function. After we trained models, we finetuned them and applied the dynamic evaluation [16]. For fair comparison, the experimental conditions, such as unit sizes, dropout rates, initialization, and the optimization method were the same as in the previous studies [23, 34, 16] except for the number of epochs by using their codes.[3] We set the epochs to be twice as large as the original epochs used in [23] since the losses did not converge in the original epochs. In addition, we trained each model with various random seeds and evaluated the average and standard deviation of validation and test perplexities for each method. The detailed conditions and the results at training and finetuning steps are provided in the supplementary material.

## 4.2 Experimental results

Validation perplexities and test perplexities of PTB and WT2 modeling are listed in Tabs. 1 and 2. Note that we confirmed these results are statistically different by pair-wise t-test (5 % of p-value). Table 1 shows that the sigmoid-based function achieved the lowest perplexities among output activation functions on PTB. However, the sigmoid-based function did not outperform softmax on WT2. This is because sigmoid is bounded above by one, $\sigma(\cdot) \leq 1$, and it may restrict the representational power. As a result, the sigmoid based function did not perform well on the large dataset. On the other hand, sigsoftmax achieved lower perplexities than softmax on PTB and achieves the lowest perplexities on WT2. Furthermore, between mixture models, MoSS achieved lower perplexities than MoS. Even though we trained and finetuned models under the conditions that are highly optimized for softmax and MoS in [23, 34], sigsoftmax and MoSS outperformed softmax and MoS, respectively. Therefore, we conclude that sigsoftmax outperforms softmax as an activation function.

## 4.3 Evaluation of linear independence

In this section, we evaluate linear independence of output vectors of each function. First, we applied whole test data to the finetuned models and obtained log-output $\log(P_{\boldsymbol{\theta}}(\boldsymbol{y}_t|\boldsymbol{x}_t))$, e.g., log-softmax, at each time. Next, we made the matrices $\hat{\boldsymbol{A}}$ as $\hat{\boldsymbol{A}} = [\log(P_{\boldsymbol{\theta}}(\boldsymbol{y}_1|\boldsymbol{x}_1)), \ldots, \log(P_{\boldsymbol{\theta}}(\boldsymbol{y}_T|\boldsymbol{x}_T))] \in \boldsymbol{R}^{M \times T}$ where $T$ is the number of tokens of test data. $M$ and $T$ were respectively 10,000 and 82,430 on the PTB test set and 33,278 and 245,570 on the WT2 test set. Finally, we examined the rank of $\hat{\boldsymbol{A}}$ since

`https://github.com/benkrause/dynamic-evaluation; https://github.com/zihangdai/mos`

Table 3: The number of linearly independent log-output vectors on test datasets: Ranks of $\hat{A}$.

|     | Softmax | $g$: ReLU | $g$: Sigmoid | Sigsoftmax | MoS | MoSS |
|-----|---------|-----------|--------------|------------|-----|------|
| PTB | 402 | 8243 | 1304 | 4640 | 9980 | 9986 |
| WT2 | 402 | 31400 | 463 | 5465 | 12093 | 19834 |

the rank of the matrix is $N$ if the matrix is composed of $N$ linearly independent vectors. Note that the numerical approaches for computing ranks have roundoff error, and we used the threshold used in [29, 34] to detect the ranks. The ranks of $\hat{A}$ are listed in Tab. 3. The calculated singular values for detecting ranks are presented in the supplementary material.

We can see that log-softmax output vectors have 402 linearly independent vectors. In the experiments, the number of hidden units is set to 400, and we used a bias vector in the output layer. As a result, the dimension of the input space $S$ was at most 401, and log-softmax output vectors are theoretically at most 402 linearly independent vectors from Theorem 2. Therefore, we confirmed that the range of log-softmax is a subset of the $d + 1$ dimensional vector space. On the other hand, the number of linearly independent output vectors of sigsoftmax, ReLU and sigmoid-based functions are not bounded by 402. Therefore, sigsoftmax, ReLU and sigmoid-based functions can break the softmax bottleneck. The ranks of the ReLU-based function are larger than the other activation functions. However, the ReLU-based function is numerically unstable as mentioned in Sec. 3.2. As a result, it was not trained well as shown in Tabs. 1 and 2. MoSS has more linearly independent output vectors than MoS. Therefore, MoSS may have more representational power than MoS.

## 5 Conclusion

In this paper, we investigated the range of log-softmax and identified the cause of the softmax bottleneck. We proposed sigsoftmax, which can break the softmax bottleneck and has more representational power than softmax without additional parameters. Experiments on language modeling demonstrated that sigsoftmax outperformed softmax. Since sigsoftmax has the desirable properties for output activation functions, it has the potential to replace softmax in many applications. Breaking the softmax bottleneck is the necessary conditions in order to fit the model to the true distribution. In our future work, we will investigate the sufficient conditions in order to fit the model to the distribution.

## Footnotes

[1] If neural networks have the universal approximation property, $\boldsymbol{h}$ can be an arbitrary vector in $\boldsymbol{R}^d$. If not, the input space is a subset of a $d$ dimensional vector space, and the range of log-softmax is still a subset of a $d+1$ dimensional vector space. When $\text{rank}(\boldsymbol{W}) < d$, we can examine the range of log-softmax in the same way by replacing $d$ with $\text{rank}(\boldsymbol{W})$. If a bias is used in the output layer, the dimension of $S$ can be $d+1$.

[2] Even though $k'^{(d)}$ is extremely large, the input vector is the element of the input space $S$.

[3]`https://github.com/salesforce/awd-lstm-lm` (Note that Merity et al. [23] further tuned some hyper-parameters to obtain results better than those in the original paper in their code.);

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
