[Supplementary Material]

# Supplementary Material for Sigsoftmax: Reanalysis of the Softmax Bottleneck

## A Proofs of theorems

In this section, we provide the proofs of theorems that are not provided in the paper.

**Theorem 3.** *Let $S \subseteq \mathbf{R}^M$ be the d dimensional vector space and $\mathbf{z} \in S$ be input of log-sigsoftmax, some range of log-sigsoftmax $\{\log(\mathbf{f}(\mathbf{z}))|\mathbf{z} \in S\}$ is not a subset of a $d+1$ dimensional vector space.*

*Proof.* We prove this by contradiction. If Theorem 3 does not hold, every range of log-sigsoftmax $\{\log(\mathbf{f}(\mathbf{z}))|\mathbf{z} \in S\}$ is a subset of a $d+1$ dimensional vector space. When we provide a counterexample of this statement, we prove Theorem 3 since this statement is the negation of Theorem 3. The counter example is the case in which $S$ is the one dimensional vector space (i.e., $d = 1$), $S = \{k\mathbf{u}|k \in R\}$ and $\mathbf{u} = [1,2,0]^T$. Under the above condition, from Definition 1 in the paper, outputs of log-sigsoftmax are as follows:

$$\log(\mathbf{f}(\mathbf{z})) = \begin{bmatrix} 2k \\ 4k \\ 0 \end{bmatrix} - \begin{bmatrix} \log(1+\exp(k)) \\ \log(1+\exp(2k)) \\ \log(2) \end{bmatrix} - \log\left(\tfrac{1+2\sigma(k)\exp(k)+2\sigma(2k)\exp(2k)}{2}\right)\mathbf{1}. \tag{1}$$

From $S = \{k\mathbf{u}|k \in R\}$, we choose three inputs $\mathbf{z}_1 = [0,0,0]^T$, $\mathbf{z}_2 = [1,2,0]^T$, $\mathbf{z}_3 = [-1,-2,0]^T$ and investigate the outputs. The outputs of log-sigsoftmax are as follows:

$$\log(\mathbf{f}(\mathbf{z}_1)) = -\log(3)\mathbf{1}, \tag{2}$$

$$\log(\mathbf{f}(\mathbf{z}_2)) = \begin{bmatrix} 2-\log(1+\exp(1)) \\ 4-\log(1+\exp(2)) \\ -\log(2) \end{bmatrix} - \log\left(\tfrac{1+2\sigma(1)\exp(1)+2\sigma(2)\exp(2)}{2}\right)\mathbf{1}, \tag{3}$$

$$\log(\mathbf{f}(\mathbf{z}_3)) = \begin{bmatrix} -2-\log(1+\exp(-1)) \\ -4-\log(1+\exp(-2)) \\ -\log(2) \end{bmatrix} - \log\left(\tfrac{1+2\sigma(-1)\exp(-1)+2\sigma(-2)\exp(-2)}{2}\right)\mathbf{1}. \tag{4}$$

To evaluate linear independence, we examine the solution of the $\alpha_1\log(\mathbf{f}(\mathbf{z}_1)) + \alpha_2\log(\mathbf{f}(\mathbf{z}_2)) + \alpha_3\log(\mathbf{f}(\mathbf{z}_3)) = \mathbf{0}$. If its solution is only $\alpha_1 = \alpha_2 = \alpha_3 = 0$, $\log(\mathbf{f}(\mathbf{z}_1))$, $\log(\mathbf{f}(\mathbf{z}_2))$, and $\log(\mathbf{f}(\mathbf{z}_3))$ are linearly independent. Each element of $\alpha_1\log(\mathbf{f}(\mathbf{z}_1))+\alpha_2\log(\mathbf{f}(\mathbf{z}_2))+\alpha_3\log(\mathbf{f}(\mathbf{z}_3)) = \mathbf{0}$ becomes the following equations:

$$
\begin{cases}
\begin{aligned}
&-\alpha_1\log(3) + \alpha_2\{2 - \log(1+\exp(1)) - \log(\tfrac{1+2\sigma(1)\exp(1)+2\sigma(2)\exp(2)}{2})\} \\
&+\alpha_3\{-2 - \log(1+\exp(-1)) - \log(\tfrac{1+2\sigma(-1)\exp(-1)+2\sigma(-2)\exp(-2)}{2})\} = 0, \quad (5)\\
&-\alpha_1\log(3) + \alpha_2\{4 - \log(1+\exp(2)) - \log(\tfrac{1+2\sigma(1)\exp(1)+2\sigma(2)\exp(2)}{2})\} \\
&+\alpha_3\{-4 - \log(1+\exp(-2)) - \log(\tfrac{1+2\sigma(-1)\exp(-1)+2\sigma(-2)\exp(-2)}{2})\} = 0, \quad (6)\\
&-\alpha_1\log(3) + \alpha_2\{-\log(2) - \log(\tfrac{1+2\sigma(1)\exp(1)+2\sigma(2)\exp(2)}{2})\} \\
&+\alpha_3\{-\log(2) - \log(\tfrac{1+2\sigma(-1)\exp(-1)+2\sigma(-2)\exp(-2)}{2})\} = 0. \quad (7)
\end{aligned}
\end{cases}
$$

From Eq. (7), we have

$$\alpha_1 = \frac{\alpha_2}{\log(3)}\{-\log(2) - \log(\frac{1+2\sigma(1)\exp(1)+2\sigma(2)\exp(2)}{2})\}$$

$$+ \frac{\alpha_3}{\log(3)}\{-\log(2) - \log(\frac{1+2\sigma(-1)\exp(-1)+2\sigma(-2)\exp(-2)}{2})\}. \tag{8}$$

Substituting Eq. (8) for Eq. (5) and Eq. (6), we have

$$
\begin{cases}
\alpha_2\{2-\log(1+\exp(1))+\log(2)\} + \alpha_3\{-2-\log(1+\exp(-1))+\log(2)\} = 0, & (9)\\
\alpha_2\{4-\log(1+\exp(2))+\log(2)\} + \alpha_3\{-4-\log(1+\exp(-2))+\log(2)\} = 0. & (10)
\end{cases}
$$

From Eq. (9), we have

$$\alpha_2 = \alpha_3 \frac{\{2+\log(1+\exp(-1))-\log(2)\}}{\{2-\log(1+\exp(1))+\log(2)\}}, \tag{11}$$

and substituting Eq. (11) for Eq. (10), we have

$$\alpha_3 \left[ \frac{\{2 + \log(1 + \exp(-1)) - \log(2)\}\{4 - \log(1 + \exp(2)) + \log(2)\}}{\{2 - \log(1 + \exp(1)) + \log(2)\}} + \{-4 - \log(1 + \exp(-2)) + \log(2)\} \right] = 0. \tag{12}$$

The solution of Eq. (12) is only $\alpha_3 = 0$, and thus, $\alpha_1 = \alpha_2 = \alpha_3 = 0$. We have $\alpha_1 \log(\boldsymbol{f}(\boldsymbol{z}_1)) + \alpha_2 \log(\boldsymbol{f}(\boldsymbol{z}_2) + \alpha_3 \log(\boldsymbol{f}(\boldsymbol{z}_3) = \mathbf{0}$ if and only if $\alpha_1 = \alpha_2 = \alpha_3 = 0$. Therefore, $\log(\boldsymbol{f}(\boldsymbol{z}_1))$, $\log(\boldsymbol{f}(\boldsymbol{z}_2))$ and $\log(\boldsymbol{f}(\boldsymbol{z}_3))$ are linearly independent, i.e., output vectors can be three linearly independent vectors even though $d + 1 = 2$. Therefore, the output of log-sigsoftmax can be greater than $d + 1$ linearly independent vectors, and thus, the range of log-sigsoftmax is not a subset of $d + 1$ dimension vector space. This contradicts the statement that every range of log-sigsoftmax $\{\log(\boldsymbol{f}(\boldsymbol{z})) | \boldsymbol{z} \in S\}$ is a subset of a $d + 1$ dimensional vector space. As a result, some range of log-sigsoftmax is not a subset of a $d + 1$ dimensional vector space. $\qquad \square$

**Theorem 5.** *Sigsoftmax has the following properties:*

1. *Nonlinearity of* $\log(\boldsymbol{g}(\boldsymbol{z}))$*:* $\log(\boldsymbol{g}(\boldsymbol{z})) = 2\boldsymbol{z} - \log(\mathbf{1} + \exp(\boldsymbol{z}))$.

2. *Numerically stable:* $\frac{\partial \log[\boldsymbol{f}(\boldsymbol{z})]_i}{\partial z_j} = \begin{cases} (1 - [\boldsymbol{f}(\boldsymbol{z})]_j)(2 - \sigma(z_j)) & i = j, \\ -[\boldsymbol{f}(\boldsymbol{z})]_j (2 - \sigma(z_j)) & i \neq j. \end{cases}$

3. *Non-negative:* $[\boldsymbol{g}(\boldsymbol{z})]_i = \exp(z_i)\sigma(z_i) \geq 0$.

4. *Monotonically increasing:* $z_1 \leq z_2 \Rightarrow \exp(z_1)\sigma(z_1) \leq \exp(z_2)\sigma(z_2)$.

*Proof.* First, we have $\log(\boldsymbol{g}(\boldsymbol{z})) = 2\boldsymbol{z} - \log(\mathbf{1} + \exp(\boldsymbol{z}))$ since $[\boldsymbol{g}(\boldsymbol{z})]_i = \exp(z_i)\sigma(z_i) = \frac{\exp(z_i)}{1 + \exp(-z_i)} = \frac{\exp(2z_i)}{1 + \exp(z_i)}$. $\log(1 + \exp(z))$ is softplus and is a nonlinear function. Therefore, $\log(\boldsymbol{g}(\boldsymbol{z}))$ is a nonlinear function. Second, since we have $\frac{d \exp(z)}{dz} = \exp(z)$ and $\frac{d\sigma(z)}{dz} = \sigma(z)(1 - \sigma(z))$, we have

$$\frac{\partial \log[\boldsymbol{f}(\boldsymbol{z})]_i}{\partial z_j} = \frac{1}{[\boldsymbol{f}(\boldsymbol{z})]_i} \frac{\partial [\boldsymbol{f}(\boldsymbol{z})]_i}{\partial z_j},$$

$$= \frac{1}{[\boldsymbol{f}(\boldsymbol{z})]_i} \left\{ \frac{1}{\sum_{m=1}^{M} \exp(z_m)\sigma(z_m)} \frac{\partial \exp(z_i)\sigma(z_i)}{\partial z_j} - \frac{[\boldsymbol{f}(\boldsymbol{z})]_i}{\sum_{m=1}^{M} \exp(z_m)\sigma(z_m)} \frac{\partial \sum_{m=1}^{M} \exp(z_m)\sigma(z_m)}{\partial z_j} \right\},$$

$$= \begin{cases} (1 - [\boldsymbol{f}(\boldsymbol{z})]_j)(2 - \sigma(z_j)) & i = j, \\ -[\boldsymbol{f}(\boldsymbol{z})]_j (2 - \sigma(z_j)) & i \neq j. \end{cases}$$

Third, since $\exp(z) \geq 0$ and $\sigma(z) \geq 0$, we have $\exp(z)\sigma(z) \geq 0$. Finally, the derivative of $\exp(z)\sigma(z)$ is $\frac{d \exp(z)\sigma(z)}{dz} = \frac{\exp(z) + 2}{(1 + \exp(-z))^2} \geq 0$ for all $z$, and thus, $\exp(z)\sigma(z)$ is monotonically increasing. $\qquad \square$

# B  Properties of output functions composed of ReLU and sigmoid

In this section, we investigate properties of output functions composed of ReLU and sigmoid.

## B.1  ReLU-based output function

A ReLU-based output function is given as

$$[\boldsymbol{f}(\boldsymbol{z})]_i = \frac{\text{ReLU}(z_i)}{\sum_{m=1}^{M} \text{ReLU}(z_m)}. \tag{13}$$

The ReLU-based function does not satisfy all the desirable properties as follows:

1. Nonlinearity of $\log(\boldsymbol{g}(\boldsymbol{z}))$: The logarithm of ReLU is as follows:

$$[\log(\text{ReLU}(\boldsymbol{z}))]_i = \begin{cases} \log(z_i) & \text{if } z_i > 0, \\ -\infty & \text{if } z_i \leq 0. \end{cases}$$

This function is obviously nonlinear.

2. Numerically unstable:

$$\frac{\partial \log[\boldsymbol{f}(\boldsymbol{z})]_i}{\partial z_j} = \begin{cases} \frac{1}{\mathrm{ReLU}(z_i)}\frac{\partial \mathrm{ReLU}(z_i)}{\partial z_j} - \frac{1}{\sum_{m=1}^{M}\mathrm{ReLU}(z_m)}\frac{\partial \mathrm{ReLU}(z_i)}{\partial z_j} & i = j, \\ -\frac{1}{\sum_{m=1}^{M}\mathrm{ReLU}(z_m)}\frac{\partial \mathrm{ReLU}(z_j)}{\partial z_j} & i \neq j. \end{cases} \tag{14}$$

We can see that the derivative of a ReLU-based function has the division by $\mathrm{ReLU}(z_i)$. Since $\mathrm{ReLU}(z_i)$ can be close to zero, the calculation of gradient is numerically unstable.
3. Non-negative: $[\boldsymbol{g}(\boldsymbol{z})]_i = \max(z_i, 0)$ is obviously greater than or equal to 0.
4. Monotonically increasing: $z_1 \leq z_2 \Rightarrow \max(z_1, 0) \leq \max(z_2, 0)$ since the derivative of ReLU is always greater than or equal to 0.

From the above, the ReLU-based function is numerically unstable. Therefore, in the experiment, we use the following function:

$$[\boldsymbol{f}(\boldsymbol{z})]_i = \frac{\mathrm{ReLU}(z_i) + \varepsilon}{\sum_{m=1}^{M}\mathrm{ReLU}(z_m) + \varepsilon}, \tag{15}$$

where $\varepsilon$ is the hyper parameter of small value. In the experiment, we used $\varepsilon = 10^{-8}$.

## B.2 Sigmoid-based output function

A sigmoid-based output function is given as

$$[\boldsymbol{f}(\boldsymbol{z})]_i = \frac{\sigma(z_i)}{\sum_{m=1}^{M}\sigma(z_m)}. \tag{16}$$

The sigmoid-based function satisfies the desirable properties as follows:

1. Nonlinearity of $\log(\boldsymbol{g}(\boldsymbol{z}))$: The logarithm of sigmoid is as follows:

$$\log(\sigma(\boldsymbol{z})) = \boldsymbol{z} - \log(\boldsymbol{1} + \exp(\boldsymbol{z})).$$

This function is obviously nonlinear.
2. Numerically stable:

$$\frac{\partial \log[\boldsymbol{f}(\boldsymbol{z})]_i}{\partial z_j} = \begin{cases} (1 - [\boldsymbol{f}(\boldsymbol{z})]_j)(1 - \sigma(z_j)) & i = j, \\ -[\boldsymbol{f}(\boldsymbol{z})]_j(1 - \sigma(z_j)) & i \neq j. \end{cases}$$

We can see that this function does not have the division. Therefore, the calculation of gradient is numerically stable.
3. Non-negative: We have $[\boldsymbol{g}(\boldsymbol{z})]_i = \sigma(z_i) \geq 0$. However, sigmoid is also bounded by 1. This may be the cause of the limitation of representation capacity.
4. Monotonically increasing: $z_1 \leq z_2 \Rightarrow \sigma(z_1) \leq \sigma(z_2)$ since the derivative of sigmoid $\sigma(z)(1 - \sigma(z))$ is greater than or equal to 0.

# C  Detailed experimental conditions and results

## C.1  Experimental conditions

### C.1.1  Conditions for activation functions

For comparing softmax, sigsoftmax, the ReLU-based function, and the sigmoid-based function, we trained a three-layer LSTM by following [5]. We used the codes provided by [5, 3].[1] Merity et al. [5] further tuned some hyper parameters to obtain results better than those in the original paper [5] in their code. For fair comparison, we only changed the code of [5] as (i) replacing softmax with sigsoftmax, the ReLU-based function and sigmoid-based function, (ii) using various random seeds, and (iii) using the epochs twice as large as the original epochs in [5]. The ReLU-based function is defined by Eq. (15), and we used $\varepsilon = 10^{-8}$.

https://github.com/benkrause/dynamic-evaluation

We used the experimental conditions of [5]. The number of units of LSTM layers was set to 1150, and the embedding size was 400. The embedding layer was tied [6]. Weight matrices were initialized with a uniform distribution $U(-0.1, 0.1)$ for the embedding layer, and all other weights were initialized with $U(-\frac{1}{\sqrt{H}}, \frac{1}{\sqrt{H}})$ where $H$ is the number of hidden units.

All models were trained by a non-monotonically triggered variant of averaged SGD (NT-ASGD) [5] with the learning rate of 30, and we carried out gradient clipping with the threshold of 0.25. We used dropout connect [5], and dropout rates on the word vectors, on the output between LSTM layers, on the output of the final LSTM layer, and on the embedding layer were set to (0.4,0.25,0.4,0.1) on PTB, and (0.65,0.2,0.4,0.1) on WT2. Batch sizes were set to 20 on PTB, and 80 on WT2. Numbers of training epochs were set to 1000 on PTB and 1500 on WT2. After training, we ran ASGD as a fine-tuning step until the stopping criterion was met. We used a random backpropagation through time (BPTT) length which is $\mathcal{N}(70, 5)$ with probability 0.95 and $\mathcal{N}(35, 5)$ with probability 0.05. We applied activation regularization (AR) and temporal activation regularization (TAR) to the output of the final RNN layer. Their scaling coefficients were 2 and 1, respectively.

After the fine-tuning step, we used dynamic evaluation [3]. In this step, we used grid-search for hyper parameter tuning provided by [3]. The learning rate $\eta$ was tuned in $[3 \times 10^{-5}, 4 \times 10^{-5}, 5 \times 10^{-5}, 6 \times 10^{-5}, 7 \times 10^{-5}, 1 \times 10^{-4}]$, and decay rate $\lambda$ was tuned in $[1 \times 10^{-3}, 2 \times 10^{-3}, 3 \times 10^{-3}, 5 \times 10^{-3}]$. $\epsilon$ was set to 0.001, and batch size was set to 100.

We applied the above procedure (training and finetuning each model, applying dynamic evaluation) 10 times, and evaluated the average of minimum validation perplexities and the average of test perplexities.

### C.1.2    Conditions for mixture models; MoS with MoSS

We trained a three-layer LSTM by following [7] to compare MoSS with MoS. We also used the codes provided by [7],[2] and only changed the code as (i) replacing MoSS with MoS, (ii) using various random seeds. After we trained the models, we finetuned them and applied dynamic evaluation to the finetuned models.

We used the experimental conditions of [7]. In this experiments, the numbers of units of three LSTM layers were set to [960, 960, 620], and embedding size was 280 on PTB. On WT2, the numbers of units of LSTM layers were set to [1150, 1150, 650], and embedding size was 300. The number of mixture was set to 15 on both datasets. In the same way as the above experimental conditions, weight matrices were initialized with a uniform distribution $U(-0.1, 0.1)$ for the embedding layer, and all other weights were initialized with $U(-\frac{1}{\sqrt{H}}, \frac{1}{\sqrt{H}})$. On both datasets, we used word level variational drop out with the rate of 0.10, recurrent weight dropout with rate of 0.5, and context vector level variational drop out with the rate of 0.30. In addition, embedding level variational dropout with the rate of 0.55 and hidden level variational dropout with the rate of 0.225 were used on PTB. On WT2, we used embedding level variational drop out with the rate of 0.40 and hidden level variational drop out with the rate of 0.225. The optimization method was the same as in the previous section.

At the dynamic evaluation step, the learning rate was set to 0.002 and batchsize was set to 100 on both datasets. On PTB, $\epsilon$ was set to 0.001 and decay rate $\lambda$ was set to 0.075. On WT2, we set $\epsilon$ to 0.002 and decay rate $\lambda$ to 0.02. All the above conditions were the same as those in [7].

We applied the above procedure five times and evaluated the average of minimum validation perplexities and the average of test perplexities.

## C.2    Results

Tables. 1 and 2 list the validation and test perplexities after the training step, fine-tuning step, and dynamic evaluation. We can see that the validation and test perplexities of sigsoftmax are significantly reduced by the dynamic evaluation. Since the dynamic evaluation adapts models to recent sequences, these results imply that the high expressive power of sigsoftmax enabled the model to more flexibly adapt to the validation and test data in the dynamic evaluation. In addition, under the conditions tuned for softmax in [5], the sigsoftmax-based model might have slightly overfitted to the training data due to the high expressive power. We observed that training perplexities of sigsoftmax are smaller

Table 1: Results of the language modeling experiment on PTB. Valid. means the validation perplexity, and dynamic eval. means dynamic evaluation [3].

|  | Softmax | $g$:ReLU | $g$: Sigmoid | Sigsoftmax | MoS | MoSS |
|---|---|---|---|---|---|---|
| Valid. w/o finetune | 61.1 ±0.4 | $(1.85\pm0.2)\times10^3$ | 60.7 ±0.2 | 61.0 ±0.2 | 58.4±0.2 | 58.4±0.3 |
| Test w/o finetune | 58.8 ±0.4 | $(1.54\pm0.2)\times10^3$ | 58.5 ±0.2 | 58.4 ±0.2 | 56.3±0.3 | 56.2±0.2 |
| Valid. | 59.2 ±0.4 | $(1.51\pm0.1)\times10^3$ | 58.7 ±0.4 | 59.2 ±0.4 | 56.8±0.2 | 56.9±0.1 |
| Test | 57.0 ±0.6 | $(1.24\pm0.08)\times10^3$ | 56.4 ±0.2 | 56.6 ±0.4 | 54.7±0.08 | 54.6±0.2 |
| Valid.+dynamic eval. | 51.2±0.5 | $(4.91\pm5)\times10^3$ | **49.2±0.4** | 49.7±0.5 | 48.6±0.2 | **48.3±0.1** |
| Test +dynamic eval. | 50.5±0.5 | $(2.78\pm8)\times10^5$ | **48.9±0.3** | 49.2±0.4 | 48.0±0.1 | **47.7±0.07** |

Table 2: Results of the language modeling experiment on WT2. Valid. means the validation perplexity, and dynamic eval. means dynamic evaluation [3].

|  | Softmax | $g$:ReLU | $g$:Sigmoid | Sigsoftmax | MoS | MoSS |
|---|---|---|---|---|---|---|
| Valid. w/o finetune | 68.0±0.2 | $(8.74\pm0.7)\times10^2$ | 72.8±0.3 | 67.8±0.1 | 65.9±0.5 | 65.1±0.2 |
| Test w/o finetune | 65.2±0.2 | $(7.97\pm0.7)\times10^2$ | 69.7±0.3 | 65.0±0.2 | 63.3±0.4 | 62.5±0.3 |
| Valid. | 67.4±0.2 | $(6.48\pm0.1)\times10^2$ | 70.8±0.1 | 67.4±0.2 | 64.0±0.3 | 63.7±0.3 |
| Test | 64.7±0.2 | $(5.93\pm0.08)\times10^2$ | 68.2±0.1 | 64.2±0.1 | 61.4±0.4 | 61.1±0.3 |
| Valid.+dynamic eval. | 45.3±0.2 | $(1.79\pm0.8)\times10^3$ | 45.7±0.1 | **44.9±0.1** | 42.5±0.1 | **42.1±0.2** |
| Test +dynamic eval. | 43.3±0.1 | $(2.30\pm2)\times10^4$ | 43.5±0.1 | **42.9±0.1** | 40.8±0.03 | **40.3±0.2** |

than those of softmax at the last epochs.[3] By tuning the hyper parameters for regularization methods such as dropout rates, sigsoftmax can achieve better performance. Figure 1 shows the singular values of $\hat{A}$ on PTB, and Figure 2 shows the top 20,000 singular values of $\hat{A}$ on WT2. On both datasets, singular values of softmax significantly decrease at 403. In addition, singular values of the ReLU-based function also significantly decrease at 8243 on PTB. On the other hand, singular values of sigsoftmax, sigmoid, MoS and MoSS smoothly decrease. Therefore, their ranks might be greater than those in Tab. 3 in the paper.

# D  Character-level language modeling

In order to evaluate the effect of breaking the softmax bottleneck, we conducted the character-level language modeling on text8 dataset [4]. On character-level language modeling, softmax bottleneck does not occur since $M$ is smaller than $d$ unlike word level language modeling. If sigsoftmax does not outperform softmax on character-level language modeling, we can see that sigsoftmax outperforms sigsoftmax on word-level language modeling due to the breaking the softmax bottleneck.

## D.1  Experimental Conditions

We trained one-layer 1024 units LSTM. We used Adam with the learning rate of 0.001. The batch size was set to 100, and we truncated back propagation with 200 steps. The output size $M$ was 27 since text8 only contains 26 lower alphabets and a space, and thus, $M < d$.

## D.2  Experimental Results

Table 3 lists the results of character-level language modeling. We can see that the results of sigsoftmax and softmax are almost the same. In this experiment, the softmax bottleneck does not occur since $M < d$. Therefore, these results indicate that the reason why sigsoftmax outperforms softmax on the word level language modeling is that sigsoftmax can break the softmax bottleneck.

Figure 1: Singular values of $\hat{A}$ on PTB test set.

Figure 2: Top 20,000 singular values of $\hat{A}$ on WT2 test set.

# E One Billion Word dataset

## E.1 Experimental Conditions

We evaluated our method on One Billion Word dataset [1]. One Billion Word is a massive dataset and contains 0.8 billion tokens and its vocabulary size is about 800 k. Due to the very large vocabulary size of One Billion Word dataset, these experiments required an efficient method. Since sigsoftmax is the alternative function of softmax, it can be used together with methods for softmax, and we applied the adaptive approach [2]. we used adaptive softmax and compared it with adaptive sigsoftmax that replaces softmax with sigsoftmax in adaptive softmax [2]. In this method, we set cutoff to [4000, 40000, 200000]. We trained two-layer 2048 units LSTMs by using SGD with the learning rate of 20 for 5 epochs. After the second epoch, we divided the learning rate by 2 for each epoch. The input word embedding size was set to 256. We set the dropout rate to 0.01, the batch size to 128, and the threshold of gradient clipping to 0.25. We unrolled the models for 20 steps for the backpropagation through time.

Table 3: Bit-per-character on text8.

|       | Softmax | Sigsoftmax |
|-------|---------|------------|
| Valid | 1.49    | 1.48       |
| Test  | 1.56    | 1.56       |

Table 4: Results of the language modeling experiment on One Billion Word dataset.

|      | Adaptive softmax | Adaptive sigsoftmax |
|------|------------------|---------------------|
| Test | 33.80±0.04       | 33.62±0.05          |

## E.2 Experimental Results

Table 4 lists test perplexities on One Billion Word dataset. This table shows that test perplexity of adaptive sigsoftmax is lower than that of adaptive softmax. Therefore, sigsoftmax can outperform softmax when we use the massive dataset and the efficient method.

## Footnotes

[1] https://github.com/salesforce/awd-lstm-lm;

[2]`https://github.com/zihangdai/mos`

[3]Note that models at the last epochs were not used for evaluation since their validation perplexities were not the lowest in the training.