[Reviews · NeurIPS 2018]

Reviewer 1



This paper proposed to increase the softmax capacity by multiplying the exponential term with sigmoid function, which does not require additional parameters. The authors have shown that in theory the proposed formulation can provide better representation capacity than the softmax function, and the results on Penn Treebank and WikiText-2 dataset also shown improvement. The improvement are also observed with mixture models. The authors also showed that the output matrix by their proposed function have higher rank than conventional function on both tasks. The proposed approach is simple with good theoretical and empirical support. On the other hand, it would be more convincing if the authors could assess their approach on larger dataset like 1B Word dataset or on other domain. The paper is clearly written.

Reviewer 2



The paper analyzes ability of the soft-max, if used as the output activation function in NN, to approximate posterior distribution. The problem is translated to the study of the rank of the matrices contating the log-probabilities computed by the analyzed activation layer. It is shown that the soft-max does not increases the rank of the input response matrix (i.e. output of the penultimate layer) by more than 1. The authors propose to replace soft-max by the so called sigsoftmax (i.e. product of sigmoid and soft-max functions). It is shown that the rank of sigsoftmax matrix is not less the rank of soft-max. The other outcome of the paper is an empirical evaluation of several activation functions on two language modeling tasks. The provided theorems studying rank of response matrices obtained by various activation functions have their values per se. However, a practical impact of the results and their relevance to the studied language modeling problem are not fully clear from the paper. In particular, the presented results characterize the rank of the matrix containing the output responses. The rank itself is clearly a proxy for true objective which is not defined in the paper but according to the experiments it seems to be the perplexity of the learned model. The question is how is the rank (i.e. quantity analyzed in the paper) related to the true objective (i.e. the perplexity). E.g. if some activation function produces response matrices with slightly higher ranks (compared to other activation function) does it imply that the corresponding NN model will have lower perplexity ? This question is not discussed in the paper. The provided thorems suggest that the proposed sigsoftmax is only slightly better than the softmax. In particular, according Theorem 4 the sigsoftmax is proved not to be worse, in terms of the rank, than the softmax. The empirical study carried on two datasets shows improvements of the sigsoftmax function over softmax by approximately 1 unit. It is not clear if this improvement is statistically relevant because: i) the number of datasets is limited and ii) the confidence of the estimate is not given (the reported STD is not a good measure of the confidence because it measures variation of the results w.r.t. network initialization according to line 268-269). A pragmatic way to increase expressiveness of NN with the soft-max (and other functions) would to simply increase dimension $d$ of the hidden (penultimate) layer. It would increase the number of parameters, however, the same happens if one uses the mixture of soft-max (MOS). It is not clear why such obvious baseline was not evaluated in the paper.

Reviewer 3



The work presents an alternative but more explicit analysis of the softmax bottleneck problem based on the dimensionality of the log-probability space that can be spanned by the model. Based on this reanalysis, desirable properties needed to break the bottleneck are discussed, followed by the proposed Sigsoftmax parameterization of a categorical distribution. Empirically, the proposed Sigsoftmax clearly improves upon the softmax formulation. Together with the previously proposed mixture structure, MoSS achieves the SOTA on two benchmark datasets for language modeling. Strength: - An elegant analysis and method motivated by the analysis - Good empirical results on standard benchmark datasets Weakness: - From the performance and empirical rank of ReLU in table 2 & 3, we can see that a higher empirical rank does not necessarily mean a better performance. Hence, it suggests a difference between producing some arbitrary log-probability matrix with a high rank and learning the "correct" log-probability matrix which is high-rank. From this perspective, no analysis/theory is provided to show that either the Sigsoftmax or the MoS has the "capacity" to model an arbitrary truth log-probability that resides in vector space up to M-dimensional. It is only shown that Sigsoftmax and MoS have a relatively better representational power than Softmax, an augment that holds for other non-linearities such as ReLU or softplus (a smooth version of ReLU with a stable grad). - The superior performance of Sigsoftmax compared to Softmax may come from other unintended sources, such as easier optimization, other than a higher rank. Hence, I will suggest the authors conduct an additional experiment on character-level language modeling like the one in the original softmax bottleneck paper [1] to eliminate such unintended benefits. Overall, this is a good paper, which can be further improved by providing the analysis suggested above. [1] Yang, Zhilin, et al. "Breaking the softmax bottleneck: A high-rank RNN language model."